# Structured deep generative models for sampling on constraint manifolds in sequential manipulation

**Joaquim Ortiz-Haro**    **Jung-Su Ha**    **Danny Driess**    **Marc Toussaint**
TU Berlin

**Abstract:** Sampling efficiently on constraint manifolds is a core problem in robotics. We propose Deep Generative Constraint Sampling (DGCS), which combines a deep generative model for sampling close to a constraint manifold with nonlinear constrained optimization to project to the constraint manifold. The generative model is conditioned on the problem instance, taking a scene image as input, and it is trained with a dataset of solutions and a novel analytic constraint term. To further improve the precision and diversity of samples, we extend the approach to exploit a factorization of the constrained problem. We evaluate our approach in two problems of robotic sequential manipulation in cluttered environments. Experimental results demonstrate that our deep generative model produces diverse and precise samples and outperforms heuristic warmstart initialization.

**Keywords:** Generative Models, Nonlinear Optimization, Constraint Graph, Robotic Sequential Manipulation

## 1  Introduction

We consider the problem of sampling points on a constraint manifold, i.e. finding diverse solutions of a nonlinear mathematical program without costs. Such problems arise throughout robotics, in particular in solving sequential manipulation problems, as it will be the focus in our application. We assume that the constraints are given in terms of piecewise differentiable equalities and inequalities. This allows us to leverage constrained optimization methods to project a randomly sampled configurations onto the manifold. However, for highly non-linear constraints and disconnected solution manifolds, local optimization methods can get trapped in local optima and fail to find a feasible solution. Further, our aim is to generate a diverse and covering set of points on the constraint manifold, and to reduce the computation time needed by the local optimization. The crucial challenge therefore is to first *sample diversely and close to the constraint manifold*, so that the local optimizer can efficiently project to solutions covering the constraint manifold.

We follow a learning-based approach, where we assume that a dataset of feasible points for various problem instances is available. Reusing precomputed data on similar problems is a promising approach [1, 2, 3]. A fundamental challenge is that the mapping between the problem instance and the feasible manifold is extremely nonlinear and discontinuous [4, 5].

Our approach, therefore, is to train a generative neural model to map a problem instance – given in terms of a scene image – to a randomized sample close to the constraint manifold for that instance. An image-based prediction of solutions to manipulation problems has recently received attention [1, 6, 7, 8, 9, 10] as it provides a fixed size parametrization (image resolution) that encodes the obstacles and the objects to be interacted, without the need to engineer problem instance features. Importantly, it is a natural way to encode a non-fixed number of objects.

Generative Adversarial Networks (GANs) [11, 12] and Variational Autoencoders (VAEs) [13] have proposed a powerful methodology for training such a generative model and shown great potential to represent complex distributions in high dimensional spaces. This work adopts the training objective and method of GANs, which is to minimize the divergence between the generative and target distributions. That is, given the training data being the diverse samples on the constraint manifold, the deep generative model is trained to produce seeds that lie close to the manifold conditioned on the current problem, represented with an image of the scene. These seeds are then used as initializa-

tion (warmstart) of a subsequent optimization procedure that projects them to the manifold to find a feasible solution.

Despite the expressive power of function approximators and recent improvements in deep generative models, they still have limitations in generating samples from a highly nonlinear and multimodal distribution. Inspired by previous works exploiting an underlying sparse structure of robotic sequential manipulation problems [14, 15, 16, 17, 18] we propose an extension of our generative framework that leverages a factorization of the constraint problem to model this multimodal distribution more efficiently. Sampling is decomposed into a sequence, where for each factor we can train a separate *conditional* generative model. Sequencing these conditional generative models increases sample efficiency of the training and the multi-modality of the distributions it can represent.

We demonstrate the system in the context of robotic sequential manipulation, where the problem is to find a sequence of feasible mode-switch configurations, e.g., robot's poses when it picks or places an object. These problems are defined by a set of continuous variables representing sequences of robot configurations, object poses, and relative transformations, with nonlinear, piece-wise differentiable constraints, e.g., for grasping, placement, collision avoidance and kinematics, which give a natural factorization of the overall problem. Sampling a diverse set of mode-switch configurations efficiently is essential as an inner module of task and motion planning, e.g. to provide waypoints for subsequent trajectory optimization.

In summary, our core contributions are the following:

- We introduce Deep Generative Constraint Sampling (DGCS), which combines deep generative models with nonlinear optimization to efficiently generate diverse solutions of constraint problems. The generative model is trained conditional to a scene image to be applicable to varying problem instances. As an application, we produce mode-switch configurations in sequential manipulation.
- We extend the standard data-based training of generative models with analytical features of the feasible manifold to achieve both diversity and sample accuracy.
- We extend our system to exploit a factorization of the constraint problem, if given, to further improve modelling of multi-modal distributions with disconnected support and reduce sample complexity.

## 2   Related Work

**Generative Models in Robotics**   Recently, deep generative models have successfully been applied in robotics, especially in settings where the problem is represented directly with images or point clouds. For example, for generating grasps of complex objects directly from images [19, 20]. In the context of motion planning [21, 22, 23] and planning [24, 25, 26], generative models are used for sampling informative collision free configurations which greatly improve running time of sampling based algorithms. In this line of research, the general aim is to train a network to directly predict partial or full solutions to a problem. However, none of these works addresses sampling on a high-dimensional constraint manifold, or predicting full multi-robot manipulation sequences. To achieve accurate sampling, we combine learned generative models with local optimization for constraint projection.

**Warmstart in Nonlinear optimization**   In robotics, nonlinear optimization is used to sample on constraint manifolds and optimize trajectories [27, 28, 29]. The goal of recent data-based approaches is to learn a warmstart to reduce the online computation time [2, 30]. In settings where the nonlinear programs can be represented with mixed integer constraints, [31, 32, 3, 33] learn an assignment to the integer constraints and run subsequent optimization. In comparison to our method, integer formulations use a discriminative model that is easier to train but their formulation is difficult to generalize to problems without a clear integer structure. Recently, [34] applies GANs for inverse kinematics, adding forward kinematics into the network training. In contrast, we use general analytical features in the cost term. Moreover, our framework is conditioned on an image representation, and scales to sequential manipulation by exploiting the factorization.

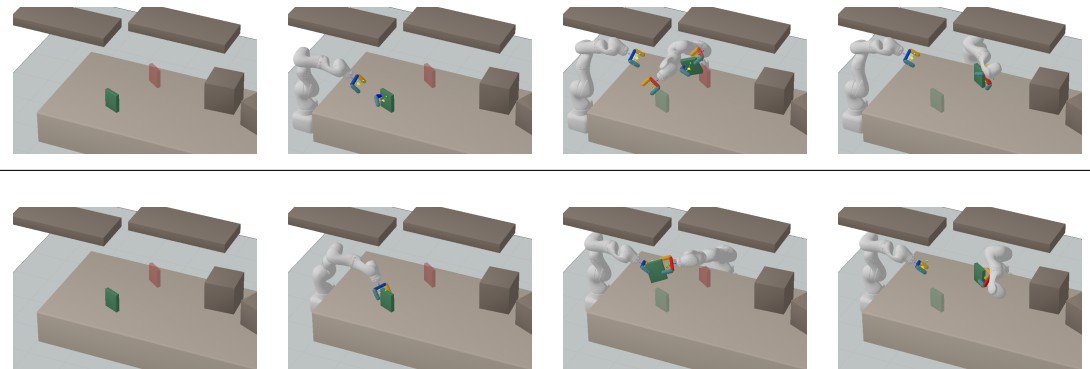

Figure 1: Sequence of mode-switches (pick - handover - place) of the *handover* problem. Our sampling framework (DGCS) combines a deep generative model $\tilde{x} \sim \mathbb{P}_\theta(\tau)$ that produces approximate samples (top row) conditioned on the scene (first column) with a nonlinear optimizer that projects them onto the constraint manifold (bottom row).

## 3   Sampling on a Constraint Manifold

The problem we address is to generate samples from a manifold $\mathcal{M}_\tau$ parametrized by a fixed dimensional (but potentially large, e.g. images) problem parameter $\tau$,

$$\mathcal{M}_\tau = \left\{ x \in \mathbb{R}^n \quad \text{s.t.} \quad \phi_{\text{eq}}(x; \tau) = 0, \ \phi_{\text{ineq}}(x; \tau) \leq 0 \right\}, \tag{1}$$

where $\phi_{\text{eq}}(x; \tau) : \mathbb{R}^n \times \mathbb{R}^m \to \mathbb{R}^{n_e}$ and $\phi_{\text{ineq}}(x; \tau) : \mathbb{R}^n \times \mathbb{R}^m \to \mathbb{R}^{n_i}$ are nonlinear vector-valued functions that are piecewise differentiable. $\tau \in \mathbb{R}^m$ represents the current problem instance and parameterizes all the constraints. In the context of robotic manipulation, $\tau$ represents all the objects in the scene (position, size, shape...), $x$ denotes robot's and objects' degrees of freedom that are subject to constraints, and $\phi_{\text{eq}}, \phi_{\text{ineq}}$ are (in)equality constraints that describe the problem's objectives such as collision avoidance, grasping, kinematic and pose constraints.

A generative model $x \sim \mathbb{P}(\tau)$ that produces feasible solutions (samples $x \in \mathcal{M}_\tau$) is built from two components (see Alg. 1): a randomized seed $\tilde{x}$ generation and a constrained optimization algorithm that projects the seed $\tilde{x}$ to the solution manifold with the optimization problem (2),

$$\min_x ||x - \tilde{x}||^2 \quad \text{s.t.} \quad \phi_{\text{eq}}(x; \tau) = 0; \ \phi_{\text{ineq}}(x; \tau) \leq 0 \tag{2}$$

In our current implementation, we solve (2) approximately, by running a nonlinear optimizer from the starting point $\tilde{x}$ on the feasibility problem: find $x$ s.t. $\phi_{\text{eq}}(x; \tau) = 0$; $\phi_{\text{ineq}}(x; \tau) \leq 0$. The initialization and internal regularization of the optimizer provides an implicit regularization with respect to $\tilde{x}$.

The projection step is a non-convex optimization problem that has no guarantees to produce a feasible sample; especially for a complex sequential manipulation problem, the optimization landscape often has substantial local optima induced by constraints, making non-linear projection prone to fail unless seeds are close to the solution manifold.

To address such difficulty, we train a seeding distribution $\mathbb{P}_\theta(\tau)$ to approximate a reference distribution of feasible solutions $\mathbb{P}_r(\tau)$, so that it can generate diverse samples close to the parametric manifold $\mathcal{M}_\tau$ on which the optimizer can easily project. An example is shown in Fig. 1.

---

**Algorithm 1** Sampling on a Constraint Manifold

1: **Input:** Problem parametrization $\tau$, number of trials $N$
2: $L = \{\}$ empty list of samples
3: **for** $i = 1, 2, ..., N$ **do**
4:     Sample $\tilde{x} \sim \mathbb{P}_\theta(\tau)$
5:     $x \leftarrow \Pi(\tilde{x})$, Project to $\mathcal{M}_\tau$ with a nonlinear optimizer (2)
6:     **if** $x$ feasible **then**
7:         append $x$ to $L$
8: **Output:** List of valid samples $L$

---

# 4 Training Deep Generative Models to Sample on Constraint Manifolds

Our deep generative model is $\tilde{x} \sim \mathbb{P}_\theta(\tau)$ with $\tilde{x} = G_\theta(z, \tau), z \sim \mathbb{P}_z$ where $\mathbb{P}_z$ is a multidimensional Gaussian distribution and $G_\theta$ is a neural network with parameters $\theta$.

In contrast to the standard application of adversarial generative models (image generation), in our setting we also have an analytical description of the support of the target distribution, namely the features $\phi(x; \tau) = [\phi_{\text{eq}}(x; \tau), \max(0, \phi_{\text{ineq}}(x; \tau))]$ that describe $\mathcal{M}_\tau$. We could consider a data-free gradient based optimization of the analytical constraint violation as

$$\min_{\tau} \mathbb{E} \; \mathbb{E}_{\tilde{x} \sim \mathbb{P}_\theta} ||\phi(\tilde{x}; \tau)||^2 . \tag{3}$$

However, this is extremely ill-posed and can converge to the "deterministic mapping" $G_\theta(z, \tau) = G_{\theta,\tau} \; \forall z$ where the model ignores the noise $z$ and loses the potential to generate a diverse distribution.

To enforce sample diversity, we regularize with respect to a reference distribution $\mathbb{P}_r(\tau)$ that is diverse and has its support on the manifold, i.e., $\mathbb{E}_{x \sim \mathbb{P}_r} ||\phi(x; \tau)||^2 = 0$. Namely, we formulate the problem as

$$\min_{\tau} \mathbb{E} \; W(\mathbb{P}_\theta(\tau), \mathbb{P}_r(\tau)) + \beta \mathbb{E}_{\tilde{x} \sim \mathbb{P}_\theta} ||\phi(\tilde{x}; \tau)||^2 , \tag{4}$$

where $W$ is the Wasserstein distance, and $\beta \in \mathbb{R}_{>0}$. Although the analytical term does not provide additional information than the support of $\mathbb{P}_r(\tau)$, as will be shown in the experiment section, its contribution is important in the context of function approximation and stochastic gradient descent in non-convex optimization.

## 4.1 Wasserstein Distance and Adversarial Formulation

The Wasserstein-1 (Earth Mover distance) between two probability distributions $\mathbb{P}_r$ and $\mathbb{P}_\theta$ (5) is defined intuitively as the cost of geometrically moving the mass from one distribution to the other under an optimal transport plan,

$$W(\mathbb{P}_r, \mathbb{P}_\theta) = \inf_{\gamma \in \Pi(\mathbb{P}_r, \mathbb{P}_\theta)} \mathbb{E}_{(x,y) \sim \gamma} ||x - y|| , \tag{5}$$

where $\Pi(\mathbb{P}_r, \mathbb{P}_\theta)$ denotes the set of all joint distributions $\gamma(x, y)$ with marginals $\mathbb{P}_r$ and $\mathbb{P}_\theta$.

Compared to other notions of distance such as Jensen-Shannon Divergence or Total Variation, adversarial generative models with Wasserstein distances are known to have good practical stability and convergence, and are less prone to mode collapse [12, 35]. Furthermore, the Wasserstein distance provides a nice interpretation in our application, because it takes into account the geometric distance between two distributions; the minimization of the geometric distance (norm in configuration space) to a distribution supported on the manifold improves the success rate of the subsequent projection procedure.

We minimize the objective function (4) with the Wasserstein GAN [12, 35] formulation. Using Kantorovich duality, the original formulation (5) is transformed into a minimax game between the critic network $D$ and the generator $G$, which are trained simultaneously with stochastic gradient descent. Specifically, the minimax problem (with our analytical feature) is:

$$\min_{G} \max_{D} \mathbb{E} \; \mathbb{E}_{x \sim \mathbb{P}_r} D(x; \tau) - \mathbb{E}_{\tilde{x} \sim \mathbb{P}_\theta} D(\tilde{x}; \tau) - \lambda \mathbb{E}_{\hat{x} \sim \mathbb{P}_{\hat{x}}} (||\nabla D(\hat{x}; \tau)|| - 1)^2 + \beta \mathbb{E}_{\tilde{x} \sim \mathbb{P}_\theta} ||\phi(\tilde{x}; \tau)||^2 , \tag{6}$$

where $\beta, \lambda \in \mathbb{R}_{>0}$ and $\hat{x}$ are samples between $x$ and $\tilde{x}$. Our reference distribution $\mathbb{P}_r(\tau)$ consists of a discrete set of solution-problem pairs $\{(x_i, \tau_i)\}$. These data are computed offline by solving a large set of problems, with a focus on getting diverse samples.

# 5 Structured Generative Model by Exploiting Factorization

The previous sections treated $x$ as a single vector variable. While the approach for training a single generative model $\tilde{x} = G_\theta(z; \tau), z \sim \mathbb{P}_z$ is powerful, we can further improve the scalability to large problems by exploiting a given factorization of $x$ and sequentially decomposing the sampling procedure into a sequence of conditional sampling operations, as in Bayesian networks [36].

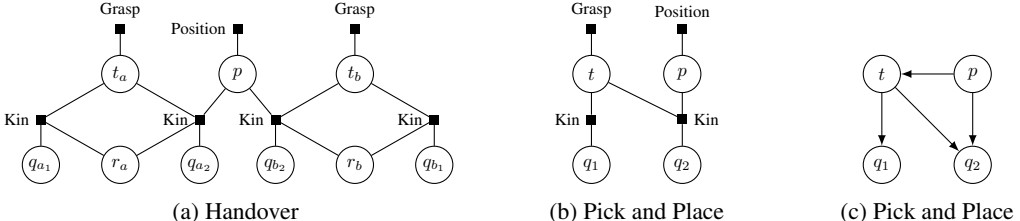

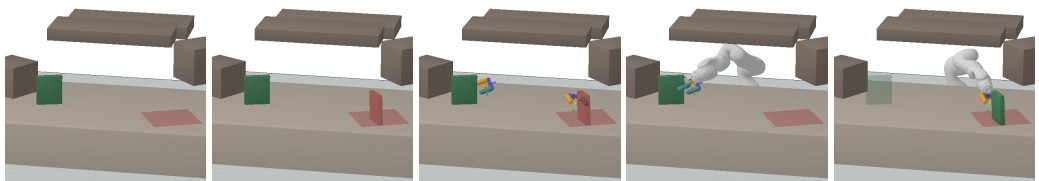

Figure 2: Constraint graphs (2b, 2a) and sampling network (2c). Additional uni-variable collision constraints are applied to $p$'s and $q$'s.

Figure 3: Sequence of learned deep generative of models for *Pick and Place* problems.

We assume the full variable $x = \{x_1, ..., x_N\}$ can be factored into $N$ vector variables. Similarly, the constraints (equalities and inequalities) $\phi(x; \tau)$ are factored into $L$ sets of constraints, $\{\phi_1, \phi_2, \ldots, \phi_L\}$ with $\quad \phi_j : \mathbb{R}^{a_j} \to \mathbb{R}^{b_j}$ where $a_j, b_j \in \mathbb{N}$ are the dimensions of the domain and co-domain of the constraint $\phi_j$, and each constraint $\phi_j$ depends only on a subset of variables.

$$x = \{x_1, \ldots, x_N\}, \quad \phi(x; \tau) = \{\phi_1, \ldots, \phi_L\} . \tag{7}$$

The set of variables and constraints now defines a factored mathematical program without costs, also called constraint graph. Such factorization naturally arise in many applications, where each variable has some semantic and geometric meaning. Fig. 2 shows some examples of such constraint graphs in the context of robotic sequential manipulation. Variables correspond to robot joint configurations $q \in \mathbb{R}^7$ or $\mathbb{R}^6$, mobile robots base poses $r \in SE(2)$ or $\mathbb{R}^2$, relative transformations between objects and grippers $t \in SE(3)$ and object positions $p \in SE(3)$. These variables are coupled by kinematic, collision avoidance, grasp and pose constraints. The pick-and-place graph in Fig. 2b is the fundamental block of sequential manipulation. Longer and complex tasks such as a handover (Fig. 2a) or an assembly can be represented by chaining "pick and place" with additional constraints.

## 5.1 Directed Graphical Model and Sequential Sampling

The factored structure directly implies a factorization of the joint probability distribution $\tilde{x} \sim \mathbb{P}(\tau)$ from which we want to sample. As in Bayesian networks we can sequentialize sampling if we decide on an ordering of variables that corresponds to a directed acyclic graph. Instead of learning a single $G_\theta$ to produce a full assignment with $\tilde{x} = G_\theta(z; \tau), \ z \sim \mathbb{P}_z,$, we learn a *conditional* model for each factor by using the corresponding marginal distributions of the original data, and the subset of corresponding analytical features in the constraint graph.

We illustrate the benefits of this factorization with the *Pick-and-Place* problem. The joint distribution for this problem is factorized into (we omit conditioning on $\tau$ for clarity)

$$P(p, t, q_1, q_2) = P(p) \, P(t|p) \, P(q_1|t) \, P(q_2|t, p) , \tag{8}$$

where $p$ is the placement position of the object, $t$ is the relative transformation between object and gripper and $q_1, q_2$ are the robot joint configurations at pick and place. The factorization exploits conditional independence between $(q_1, p)$ given $t$ and $(q_1, q_2)$ given $t$. We leverage this structure by training a sequence of conditional samplers $p \sim \mathbb{P}_p$, $t \sim \mathbb{P}_t(p)$, $q_1 \sim \mathbb{P}_{q_1}(t)$, $q_2 \sim \mathbb{P}_{q_2}(t, p)$, see Fig. 2c and Fig. 3. This factorization can be easily extended to longer manipulation sequences, with the ordering $p \to r \to t \to q$. See Appendix A for more examples of sampling orders and details.

As a side note, using marginal distributions of a jointly consistent dataset is necessary, as only the marginals contain useful information of whether a partial assignment will admit a full solution. For

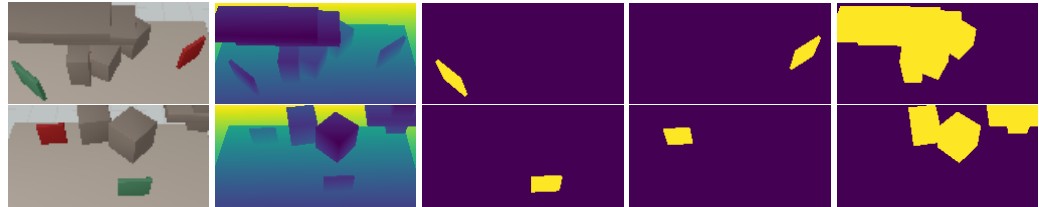

Figure 4: Problem instances in *Assembly* and *Handover*

example, when sampling $p \sim \mathbb{P}_p$, we want to fulfil the two following constraints (conditioned on the scene): a) position$(p) = 0$; and b) $\exists (t, q_1, q_2)$ s.t $(t, q_1, q_2, p)$ is feasible. The second constraint cannot be evaluated efficiently, but is modelled by the marginal distribution of the data.

### 5.2 Note on the advantage of factorization for modelling mulimodality

Generating samples from a distribution with disconnected supports with a deep generative model that receives a continuous input noise $z \sim \mathbb{P}_z$ requires infinite gradients in $G_\theta$ and can only be done approximately. In these cases, training is unstable and sensitive to hyperparameters and architecture.

We can model disconnected distributions more effectively by factoring the full joint probability as a sequence of smaller conditional modules, as is confirmed by our experimental results in Section 6.3. The sequencing still requires that each module is able to produce some degree of multimodality. However, once a module in the sequence receives a disconnected input in the form of conditioning, it can successfully produce a disconnected output. As we chain modules with the ability to generate a small amount of disconnected components given a continuous input, the number of possible disconnected components of the output grows exponentially with the number of modules in sequence.

Furthermore, from a practical perspective, training smaller modules turned out to be more effective. For instance, we observed that the analytical feature $||\phi(x; \tau)||^2$ of the joint problem can provide badly-conditioned gradients when evaluated far from the manifold. This issue is alleviated when considering only subsets of constraints and variables. In our preliminary experiments, we also evaluated GAN frameworks that have a mechanism to model disconnected distribution [37, 38] (these methods require an estimate of the number of disconnected components and a mechanism to cluster samples into components) but did not find significant improvements.

## 6 Experiments

### 6.1 Image based Problem representation

We use an image-based representation of the problem instance $\tau$ that consists of the depth image and masks. Specifically, $\tau = \{d, m_1, m_2, m_3\}$, where $d$ is the depth image, and $m_1$, $m_2$, $m_3$ are three masks that contain, respectively, information from the initial object pose, goal pose or placement region, and obstacles, see Fig. 4. In the factored approach, each generative module receives as input only the "relevant" masks, e.g. the sampler for the robot pick configuration receives a mask of the obstacles and the initial configuration, but not the goal pose.

The main strength of the image representation is that it can generalize to different object shapes and changing number of obstacles and shapes. Moreover, a depth camera is easily available, approximate masks can be computed with image segmentation techniques and it provides a good representation of sequential manipulation problems on tabletop scenarios.

### 6.2 Scenarios

We consider three different manipulation tasks that involve object manipulation with stable grasps.

- *Pick-and-Place*: A robot has to pick an object and place it on a rectangular region of the table. Fig. 3.
- *Assembly*: two mobile robots have to pick an object each and assemble them. The assembly is not unique, but modelled as a manifold with constraints on rotation and position: objects

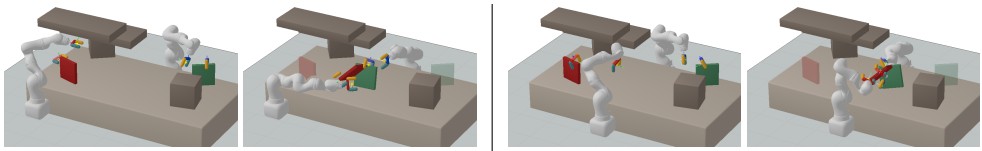

Figure 5: Samples from the deep generative model in the same instance of the *assembly* scenario. Each sample is shown with two keyframes (pick and assembly). Both seeds lead to feasible solution.

| | Seeds | | | Solutions | | |
|---|---|---|---|---|---|---|
| | Coverage | Precision | Error | Coverage | Precision | Success Rate |
| Big NN | 0.81±0.09 | 0.7±0.11 | 8.38±1.82 | 0.58±0.12 | 0.39±0.06 | 0.46 |
| Big NN + analytical | 0.79±0.12 | 0.53±0.14 | 1.21±1.35 | 0.75±0.13 | 0.41±0.13 | 0.43 |
| Structure NN | 0.6±0.08 | 0.62±0.1 | 8.09±1.35 | 0.41±0.05 | 0.44±0.06 | 0.56 |
| Structure NN + analytical | 0.57±0.08 | 0.47±0.13 | 1.46±0.79 | 0.44±0.07 | 0.28±0.08 | 0.78 |

Table 1: Ablation study in the *Pick-and-Place* scenario.

    have to be perpendicular and make stable contact with predefined faces of the cubes (similar to a T shape). Fig. 5.

- *Handover*: Two mobile robots have to collaborate to move the object from the initial to the goal position doing a handover. Fig. 1.

All these tasks are performed on a cluttered table with a varying number (between 3 and 5) of obstacles. The grasp between the gripper and the object is modelled with a two finger gripper (e.g. Franka Panda Gripper) that constrains the position and orientation. The training dataset consists of 4000 pairs of problem-solution computed offline with a user-defined sampling sequence to ensure diversity. The problems consist of 64x128x4 images (input of network) and their corresponding environment (for the analytical error term in training and nonlinear optimization).

The variability between instances is defined by the number, position and size of obstacles, the size and position of the objects and the goal configuration. The instances of the evaluation and train dataset come from the same distribution. See the Appendix A for details about the structure of the networks and implementation, and a collection of samples generated by our deep generative models.

## 6.3 Ablation Study

The *Pick-and-Place* scenario is used for an ablation study of the proposed generative model. We analyze the contribution of the factored structure (*Big NN* vs *Structure NN*) and analytical error term (+*analytical*). We evaluate precision and coverage of seed samples (output from the deep generative models) and solutions (after projection with nonlinear optimization) by generating 4000 samples for each new instance (30 in total). Results are shown in Table 1. We report the metrics:

- *Error*: the constraint violation $||\phi(\tilde{x}; \tau)||^2$ (unitless, lower is better).
- *Precision*: average nearest neighbour distance to a reference dataset. It models how close samples are to the real data (lower is better).
- *Coverage*: average nearest neighbour distance from the reference dataset to the computed samples. It describes how well the learned distribution covers the reference dataset (lower is better).

When analyzing the seeds (output of the generative model) our two contributions are essential to achieve small constraint violation (anlytical term) and good coverage (structure). Seeds from the model with structure and analytical term have higher probability of leading to a solution (success rate) and, after the projection, only samples from networks with structure provide good coverage.

## 6.4 Benchmark: Generative Model in Nonlinear optimization

The *Assembly* and *Handover* scenarios are used to compare our generative model against two baseline methods for warmstarting (seeding) nonlinear optimizers. We analyze the number of solved problems and the number of necessary optimization runs. Measuring the number of solved nonlin-

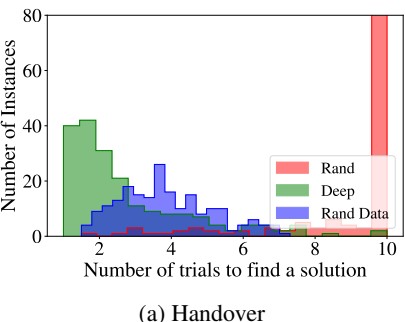

(a) Handover

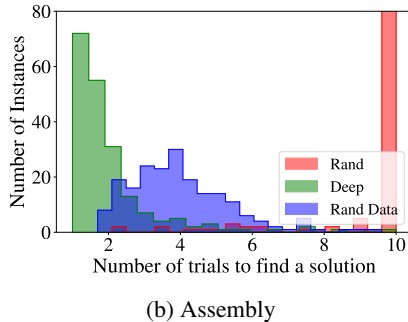

(b) Assembly

Figure 6: Histogram of the estimated number of trials necessary to solve an instance.

ear programs is an indirect way to evaluate coverage and sample quality, as both are fundamental to solve a diverse set of problems with a nonlinear optimizer and prevent convergence to infeasible points. We compare our complete model (deep generative model with structure and analytical error term), in short *deep*, with baselines:

- *Rand:* Randomized initial guess around a reference value.
- *Rand Data:* Choosing samples $x$ from the training dataset at random. The initial point is thus a feasible sample for another problem of the same family. This is actually a strong baseline, because it provides diverse informative initial seeds.

We evaluate the generative model (deep generative model + optimization) on 200 problems from the evaluation dataset. The experiments are repeated 10 times and we report mean and variance. We first report how many optimization trials (each trial has an independent starting point) are necessary to solve each of the test instances, and plot the histogram of the mean value in Fig. 6. Unsolved problems are assumed to be solved with 10 trials (maximum number trials).

In both scenarios, the proposed deep generative model outperforms the baseline warmstarts, significantly reducing the number of trials required to solve the instances: in average (across problems), from (Rand Data, 3.86±1.29) to (Deep, 2.77±1.79) in *Handover* and from (Rand Data, 3.99±1.44) (Deep, 2.07±1.38) in *Assembly*. To complete the analysis, we also show the cumulative number of problems solved as we increase the number of optimization trials in Fig. 7 of the Appendix A. The computational overhead of evaluating the neural network is small (we produce 10 samples in 8 ms with a GPU), while most of the time is spent in optimization runs that converge to infeasible points. Time spent (in seconds) to complete the benchmarks are: (Deep, 949±32), (Rand Data, 1746±76) in *Assembly*; and (Deep, 1246±26), (Rand Data, 1715±58) in *Handover*.

## 7 Conclusion

In this work, we propose Deep Generative Constraint Sampling (DGCS), a new approach to sample on a constraint manifold to tackle problems in robotic sequential manipulation. Our framework combines a deep generative sampling model, conditioned on an image based representation of the problem, and a nonlinear optimizer to project samples onto the manifold. We further extended the approach to exploit a given factorization of the problem, by training a sequence of conditional generative models rather than a full joint generator. Our empirical results confirm that the trained generative models outperform heuristic warmstart strategies. Moreover, the inclusion of analytic constraints in the training of the generative model, as well as exploiting the factorization of a given problem significantly improves the efficiency, diversity and precision of the sampling approach.

A limitation of our approach is that training requires a dataset of solutions to different instances of the same problem class. As future work, the proposed graphical structure could be exploited to provide generalization across different problem classes (i.e. different type and number of constraints and variables) by sharing and combining the sampling modules of the sequence. Our current framework combines generative sampling using a neural network with subsequent projection using constrained optimization. A promising future direction is to explor.e whether it is possible to embed the optimization algorithm as a last layer of the generative model, while keeping good coverage and multimodality.

**Acknowledgments**

Joaquim Ortiz-Haro and Danny Driess thank the International Max-Planck Research School for Intelligent Systems (IMPRS-IS) for the support. This research has been supported by the German-Israeli Foundation for Scientific Research (GIF) grant I-1491-407.6/2019 and the German Research Foundation (DFG) under Germany's Excellence Strategy – EXC 2002/1–390523135 "Science of Intelligence".

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
