# OpenReview forum: "Structured deep generative models for sampling on constraint manifolds in sequential manipulation"
_robot-learning.org/CoRL/2021/Conference — CoRL2021 Poster_

### Official Review · Reviewer_2JcL · 2021-07-19

**Originality:** Very Good
**Technical Quality:** Excellent
**Clarity Of Presentation:** Very Good
**Impact:** 4

**Recommendation:**

Strong Accept: I recommend accepting the paper and will argue for my recommendation even if other reviewers hold a different opinion.

**Summary:**

The paper presents Deep Generative Constraint Sampling (DGCS), a novel constraint manifold sampling for sequential manipulation. The proposed framework combines a deep generative sampling model -conditioned on an image-based representation of the task, and trained with a regularization using a Wasserstein distance to a reference probability distribution- and a nonlinear optimizer for sample projection onto the manifold. Moreover, the learning framework are infused with a structure based on a given Directed Graphical Models factorization of the problem, by training a sequence of conditional generative models rather a full joint generative model. The experimental results confirm that the proposed framework achieved efficiency, diversity, and precision of the sampling approach, outperforming baselines and heuristic warmstart initialization.

**Issues:**

(1) I suggest to list down the all the known limitations of the framework/method which open venues for future works/extension of this work.
(2) In Table 1, what is the unit of the Error (I assume it's not unitless and not normalized)? Does this "Error" corresponds to "Constraint Violation" metric as mentioned in section 6.3? Maybe it's worth to make the terminology consistent between these two, to make it more readable/understandable.

**Reviewer Expertise:**

Very good: Comprehensive knowledge of the area

**Strengths And Weaknesses:**

Strengths:
The paper presents an interesting view on a probabilistic technique that combines structure and learning to achieve superior results in robotic sequential manipulation tasks.

Moreover, the paper is also well-written, and relatively easy to be read.

Weaknesses:
Only a few minor points in the "Issues" section.

**Summary Of Recommendation:**

I think the paper present an interesting technique of mixing structure and learning in a probabilistically-sound manner which deserves a publication.

---

> ### Author Response · Authors · 2021-08-25
> **Response to reviewer 2JcL**
>
> Dear Reviewer, thanks for your comments and suggestions.
>
> **(Reviewer)** The paper presents an interesting view on a probabilistic technique that combines structure and learning to achieve superior results in robotic sequential manipulation tasks.
>
> **(Reply)**  Thanks!
>
> **(Reviewer)** I suggest listing down the all the known limitations of the framework/method which open venues for future works/extension of this work.
>
> **(Reply)**  Agreed. We have added a new paragraph to the conclusion to discuss the limitations of the method. Training requires a dataset of solutions to different instances of the same problem class. As future work, the proposed graphical structure could be exploited to provide generalization across different problem classes (i.e. different type and number of constraints and variables) by sharing and combining the sampling modules of the sequence.
>
> **(Reviewer)** In Table 1, what is the unit of the Error (I assume it's not unitless and not normalized)? Does this "Error" corresponds to "Constraint Violation" metric as mentioned in section 6.3? Maybe it's worth to make the terminology consistent between these two, to make it more readable/understandable.
>
> **(Reply)**  Yes, error corresponds to the constraint violation. We have unified the notation.

---

### Official Review · Reviewer_8mJB · 2021-07-23

**Originality:** Good
**Technical Quality:** Very Good
**Clarity Of Presentation:** Very Good
**Impact:** 3

**Recommendation:**

Weak Accept: I recommend accepting the paper, but will not argue for my recommendation if the majority of other reviewers have a different opinion.

**Summary:**

This paper present a generative model to sample near a constraint manifold such that they can be used within a nonlinear optimizer to handle constraints. This is applied on sequential manipulation problems with some factorizations (to capture multi-modal distributions).


**Issues:**

See weaknesses


**Reviewer Expertise:**

Good: General knowledge of the area

**Strengths And Weaknesses:**

Strengths
+ Leveraging learning and working with images to solve tasks with a hybrid (discreet + continuous) action space has a lot of promising applications where pure optimization (w/o learning) approaches fall short due to strong input assumptions and high compute costs.
+ The idea to factorize the state into modular chucks is a nice way to add structure and improve data efficiency (although hand crafting this for every new task seems like a burden).

Weaknesses
- More exposition of the big picture: In Sec 1 or somewhere before Sec 3 (that jumps straight into sampling) could benefit from grounding the problem setup a bit i.e. where does the sampling fit in the larger picture of nonlinear optimization that solves some example  manipulation task?
- Can Eq. (2) alternatively be setup as a piece-wise nonlinear optimization (wherein the constraint pieces are differentiable) where the discreet decisions are either learned or sampled with heuristics or combinatorially searched? What are the trade-offs and implications here and should these be potential baselines?
- What changes between train and test settings? The variations in each of the problem scenarios seems limited so the generalization capacity is unclear.
- The overall computational time stats are helpful. It would also be helpful to get a sense for how much computation is saved by choosing to learn the sampling vs alternatives while keeping everything else the same.

**Summary Of Recommendation:**

Overall the paper has interesting ideas about using learning as 'one' tool in a box of multiple other tools and them them together towards solving a difficult problem. The current experiments brings in concerns about scaling and generalization.

---

> ### Author Response · Authors · 2021-08-25
> **Response to Reviewer 8mJB (Part 1 of 2)**
>
> Dear reviewer, thanks for your comments and suggestions.
>
> **Note:** The response has been split into 2 parts. This is part 1.
>
> **(Reviewer)** Leveraging learning and working with images to solve tasks with a hybrid (discreet + continuous) action space has a lot of promising applications where pure optimization (w/o learning) approaches fall short due to strong input assumptions and high compute costs.
>
> **(Reply)** Thanks!
>
> We want to emphasize that there are no explicit discrete variables in our problems, as we are sampling from a nonlinear piecewise differentiable manifold. However, the continuous variables and piecewise differentiable constraints hide a strong multimodal structure, induced by the grasp and collision avoidance constraints.
>
> In the context of task and motion planning, the discrete decisions are normally the symbolic actions (i.e. deciding which robot should pick which object). The sequence of actions defines a nonlinear program, whose solution is a nonlinear manifold. The application of the proposed method is to generate efficiently diverse solutions of this nonlinear program.
>
> **(Reviewer)** More exposition of the big picture: In Sec 1 or somewhere before Sec 3 (that jumps straight into sampling) could benefit from grounding the problem setup a bit i.e. where does the sampling fit in the larger picture of nonlinear optimization that solves some example manipulation task?
>
> **(Reply)**  We agree, and we have added another sentence to the introduction, but we are constrained by space.
>
> The proposed deep generative algorithm is used to generate the mode-switch configuration of a manipulation task. Sampling a diverse set of such mode-switch configurations efficiently is essential as an inner module of task and motion planning, e.g. to provide waypoints for subsequent trajectory optimization. Considering a single solution would be insufficient, as the corresponding trajectory could be infeasible, and would therefore compromise the success of the overall solver. Instead, to robustly generate a trajectory or motion,  one should consider a diverse set of mode-switches (i.e. sampling), and optimize trajectories from all of them until a solution is found.
>
> **(Reviewer)** Can Eq. (2) alternatively be setup as a piece-wise nonlinear optimization (wherein the constraint pieces are differentiable) where the discreet decisions are either learned or sampled with heuristics or combinatorially searched? What are the trade-offs and implications here and should these be potential baselines?
>
> **(Reply)** Eq. (2) is in itself a nonlinear optimization problem with a quadratic cost and nonlinear equality and inequality constraints.
>
> Only in some special cases, this equation could be reformulated by rewriting the nonlinear constraints as a disjunction of linear constraints and integer variables while keeping the number of integer variables low. The introduction of integer variables would require mixed integer programming solvers, which are computationally much more expensive. In our experience, nonlinear optimization (without integer variables) with a good initial guess is a very good strategy for sampling such manifolds. Therefore, the focus of this work is to generate good and diverse initial guesses for the nonlinear optimizer.
>
> Related work in robotics about learning for mixed integer programming has been cited throughout the paper. We have not used them as a benchmark because these methods would require a mixed-integer structure of the constraints, which is not available in our problem formulation/setting as we are applying our framework to sample from manifolds defined by general nonlinear constraints.
>
> **(Reviewer)** What changes between train and test settings? The variations in each of the problem scenarios seems limited so the generalization capacity is unclear.
>
> **(Reply)**  The training and test datasets come from the same distribution of problems (i.e. they have been generated using the same randomized procedure, but with different random seeds). The variation between problems comes from the number, size and position of obstacles, the size and position of the manipulable objects and the goal configuration. Overall, this induces a diverse manifold of solutions for each different instance. Clarification about these topics has been added to the document.

---

> ### Author Response · Authors · 2021-08-25
> **Response to Reviewer 8mJB (Part 2 of 2)**
>
> Dear reviewer, thanks for your comments and suggestions.
>
> **Note:** The response has been split into 2 parts. This is part 2.
>
> **(Reviewer)** The overall computational time stats are helpful. It would also be helpful to get a sense for how much computation is saved by choosing to learn the sampling vs alternatives while keeping everything else the same.
>
> **(Reply)**  We totally agree, thanks. In the first submission, we have reported a time comparison between A) evaluating a sampling network and B) solving the whole problem set. In this revision,  we have added a comparison of the computational times needed by our algorithm and baselines to finish the benchmark.
>
> **(Reviewer)** The current experiments brings in concerns about scaling and generalization. **(Reviewer)** The idea to factorize the state into modular chucks is a nice way to add structure and improve data efficiency (although hand crafting this for every new task seems like a burden).
>
> **(Reply)**  Generalization and scaling remain an open question, but we would like to argue that the chosen problem parametrization (image-based) offers a good generalization to diverse shapes and obstacles.
> Also, the proposed factorization can be leveraged to scale to larger problems while keeping the sample complexity low. For choosing a sampling network for new tasks in sequential manipulation, we propose to use the general ordering: objects -> relative transformations -> robots.
>
> Apart from this general ordering, some knowledge about the problem is always desirable, especially if we want to use approximate conditional independence in the sampling network (Appendix A.2 Approximate Conditional Independence).

---

> > ### Comment · Reviewer_8mJB · 2021-09-02
> > **response to authors**
> >
> > Thank you for the replies, adding explanations and the extra timing comparisons to the draft. I have increased my score on technical quality from 'good' to 'very good'. Generalization and scaling are a still a concern, but given the contributions and potential for future work, I am leaning on the positive side and in favor of acceptance, so I'll leave my original overall score.

---

### Official Review · Reviewer_yooP · 2021-07-24

**Originality:** Good
**Technical Quality:** Very Good
**Clarity Of Presentation:** Good
**Impact:** 4

**Recommendation:**

Weak Accept: I recommend accepting the paper, but will not argue for my recommendation if the majority of other reviewers have a different opinion.

**Summary:**

The paper proposed a theory for generalized manifold constrained problems in robotics and presents a framework for generative modeling of it. The generative models are constructed with Wasserstein GAN. The authors also present a graphical model for modeling the distribution of sequential problems in robotics. Experiments are shown on three robotics manipulation tasks: Pick-and-Place, Assembly, and Handover.

**Issues:**

The authors can address the items under "Weakness" in my main review section.

**Reviewer Expertise:**

Good: General knowledge of the area

**Strengths And Weaknesses:**

Strength:

- The theory presented in this paper is novel in terms of problem formulation and solution. The theory of generative modeling on the constrained manifold is a generalization to many problems in robotics. The idea of incorporating graphical models for sequential tasks is interesting.
- The presentation is overall clear, especially with clear math formulas.
- The experiment is extensive with 3 different scenarios to show the effectiveness of the proposed method.

Weakness:

- What is the deep model used in this paper? It is not very clear. Do both RGB images and depth images used in the experiments? What deep model is used in the experiments?
- The experiment results are hard to interpret. How to connect "Average number of trials to find a solution", precision and coverage to more commonly used metrics for manipulation tasks such as success rate?
- The author can further show some failure cases and explain the reason for the failure.
- The visualizations of the scenarios are hard to interpret.
  - For example, how are the two middle images in Fig 1 connected by constraint manifold projection?
- The baseline compared against in the experiment is weak. The author should at least compare against a deterministic deep learned approach.




**Summary Of Recommendation:**

Though there are some technical details missing and stronger baseline(s) needed, the paper overall would interesting to the community. It re-formulates many problems in robotics as a generalized problem and solves it with generative modeling, which could be valuable and inspiring to the community.

---

> ### Author Response · Authors · 2021-08-25
> **Response to Reviewer yooP (Part 1 of 2)**
>
> Dear Reviewer, thanks for your comments and suggestions.
>
> **Note:** The response has been split into 2 parts. This is part 1.
>
> **(Reviewer)** The theory presented in this paper is novel in terms of problem formulation and solution. The theory of generative modeling on the constrained manifold is a generalization to many problems in robotics. The idea of incorporating graphical models for sequential tasks is interesting.
>
> **(Reply)** Thanks!
>
> **(Reviewer)** What is the deep model used in this paper? It is not very clear. Do both RGB images and depth images used in the experiments? What deep model is used in the experiments?
>
> **(Reply)** The structure of the deep model is presented in the Appendix, section A.3  We have added a cross-reference from the main text (it was missing in the original submission). In short, for each sampling module, we encode the image based representation (depth channel + masks that highlight obstacles and objects) with three convolutional layers, and then combine the encoding with the low dimensional conditioning (if any) and random gaussian noise with a fully connected network.
>
> **(Reviewer)** The experiment results are hard to interpret. How to connect "Average number of trials to find a solution", precision and coverage to more commonly used metrics for manipulation tasks such as success rate?
>
> **(Reply)**  We prefer the metric "Average Number of tries to solve a problem" rather than a "success rate", because the former captures that some problem instances are harder and that, in these cases, multiple restarts are required.  The number of restarts is directly related to the running time.
> Figure 7 shows the number of solved problems (y-axis) as we increase the number of trials (x-axis). The ratio between solved problems and total number of problems can be interpreted as the success rate.
> We have added some of these numbers in the text (average of the number of trials needed to solve a problem, and success rate with one and up to 3 trials), to help the reader understand the benchmarks.
>
> Good coverage metrics are fundamental when the sampling algorithm is used as the inner module of a motion planning or task and motion planning pipeline.
>
> **(Reviewer)** The baseline compared against in the experiment is weak.
>
> **(Reply)**  Using a dataset of solutions, i.e. randomly choosing feasible solutions of a different problem instance as warmstart for the new problem, is already a good baseline because it provides a diverse and informed initial guess  (called RAND DATA in Fig. 6 and Fig. 7). In the experiments, we observe that this baseline is already much better than the basic strategy of randomly sampling the configuration space (called RAND).
>
>
> **(Reviewer)** The author should at least compare against a deterministic deep-learned approach.
>
> **(Reply)**  The goal of this project is to generate diverse solutions of a feasibility problem (i.e samples on a constraint manifold) as opposed to a single solution. This goal cannot be achieved by a deterministic deep learning approach.
>
> The reason that we set ourselves the goal of sampling diverse solutions is that computing such a diverse set of solutions is usually required as an inner module of some larger application or as the first level of hierarchical optimization. For example, in our application of robotic sequential manipulation, the samples are later used as waypoints for trajectory optimization or motion planning. Considering a single solution would be insufficient, because the corresponding trajectory could be infeasible, and would therefore compromise the success of the overall solver.  To robustly generate a trajectory or motion,  one should consider a diverse set of mode-switches (i.e. sampling), and optimize trajectories from all of them until a solution is found.
>
> In the benchmark of Warmstart for Nonlinear optimization (Section 6.4), a deterministic function would be limited to a single trial per problem, as it can not provide diverse initial guesses. As shown in the benchmark (Figure 6 and Figure 7), neither of the benchmarks nor our method is able to solve all problem instances with a single trial. Therefore, also our evaluations show that  generating diverse initial guesses is essential for robustly tackling complex nonlinear optimization.
>
> In terms of diversity and coverage of solutions (Section 6.3 Ablation Study), our generative model (and also the baselines) are able to produce diverse samples on the solution manifold and therefore outperform a deterministic function that only produces a single solution for each problem.
>
> Moreover, supervised learning of a deterministic function with diverse samples for each problem instance could suffer from conflicting gradients during training. In the best case, a deterministic model could learn either a solution "average" or collapse to a single approximate solution for each problem.

---

> ### Author Response · Authors · 2021-08-25
> **Response to Reviewer yooP (Part 2 of 2)**
>
> Dear Reviewer, thanks for your comments and suggestions.
>
> **Note:** The response has been split into 2 parts. This is part 2.
>
> **(Reviewer)** The visualizations of the scenarios are hard to interpret. For example, how are the two middle images in Fig 1 connected by constraint manifold projection?
>
> **(Reply)**  We tried to improve the visualization of the scenarios in the main text with updated figures and captions. In figure 1, middle image (column three in the new version), the constrained manifold projection means that the two robots have to grasp the object (i.e. doing a handover). A more detailed presentation of the three scenarios is provided in the supplementary video.
>
> **(Reviewer)** The author can further show some failure cases and explain the reason for the failure.
>
> **(Reply)** In this work, we could distinguish between three type of failure cases: samples of the deep generative that are very imprecise (i.e. high constraint violation), samples of the generative model that are not diverse in a given instance, and samples of the generative model that, when used as warmstart, do not lead to a feasible solution.
> Given the space constraints of the paper, we would like to point to the original collection of samples in Appendix A.4 (also shown in the supplementary video) for a graphical description of the samples of the deep generative model. Our intention is that these images provide a good visual intuition of the quality and multimodality of the samples and the type of precision errors, e.g. inexact grasping, collisions or imprecise robot configurations.

---

> ### Comment · Reviewer_yooP · 2021-09-03
> **Comments after rebuttal**
>
> Thanks for the rebuttal from the authors. I keep my original rating and vote for acceptance.

---

### Author Response · Authors · 2021-08-25
**Response to Reviewers**

We  thank  the  reviewers  for  their  detailed  comments!   Based  on  the  reviews, we have uploaded a new version of the paper with the following modifications (highlighted in blue in the revision):

- Updated figures for better visualization of the scenarios (Figure 1: We also display the environment in the first column, and clearer images. Figure 3: We display the environment in the first column, clearer images. Figure 5: clearer images.  Figure 6 of first submission: Partitioned into new Figure 6 in Section 6.4  and new Figure 7  in Appendix A.1).
- Clarification about train/evaluation dataset and variability between instances
- Better contextualization for sampling mode-switch configurations
- Typo corrections
- New paragraph about limitations and future work
- Report computational times of the benchmark
- Better explanation of benchmark results, reporting average metrics
- Cross references to the supplementary material

We will also add a comment to each review to respond individually to the points raised by the reviewers.

---

### Meta-Review · Area_Chair_AbZR · 2021-09-04

**Recommendation:** Accept (Poster)
**Confidence:** 2

**Metareview:**

*I am the emergency meta-reviewer assigned to this paper, which is why there wasn't a meta-review provided before the rebuttal.*

The reviewers have unanimously voted to accept the paper, so that is my recommendation as well.

Pros:
- The authors propose using a generative model to improve the sample efficiency of sampling poses for non-linear optimization with respect to robot poses, such that the trajectory over poses solves the task ("the constraint manifold").

Cons:
- a formal definition of constraint manifolds is provided in the first sentence of the paper, but it is not clear what specific kind of robotic constraint is formulated as a constraint manifold until L99 of the paper. Overall I found the paper difficult to understand.

---

### Decision · Program_Chairs · 2021-09-13

**Decision:**

Accept (Poster)

**Comment:**

*I am the emergency meta-reviewer assigned to this paper, which is why there wasn't a meta-review provided before the rebuttal.*

The reviewers have unanimously voted to accept the paper, so that is my recommendation as well.

Pros:
- The authors propose using a generative model to improve the sample efficiency of sampling poses for non-linear optimization with respect to robot poses, such that the trajectory over poses solves the task ("the constraint manifold").

Cons:
- a formal definition of constraint manifolds is provided in the first sentence of the paper, but it is not clear what specific kind of robotic constraint is formulated as a constraint manifold until L99 of the paper. Overall I found the paper difficult to understand.